# Flash Drought Response to Precipitation and Atmospheric Evaporative Demand in Spain

**Iván Noguera [1], Fernando Domínguez-Castro [2,3] and Sergio M. Vicente-Serrano [1,\***

[1]  Instituto Pirenaico de Ecología, Consejo Superior de Investigaciones Científicas (IPE–CSIC),
    E-50059 Zaragoza, Spain; i.noguera@ipe.cisc.es
[2]  Department of Geography, University of Zaragoza, 50009 Zaragoza, Spain; fdominguez@unizar.es
[3]  ARAID Foundation, 50018 Zaragoza, Spain
[*]  Correspondence: svicen@ipe.csic.es

**Abstract:** Flash drought is the result of strong precipitation deficits and/or anomalous increases in atmospheric evaporative demand (AED), which triggers a rapid decline in soil moisture and stresses vegetation over short periods of time. However, little is known about the role of precipitation and AED in the development of flash droughts. For this paper, we compared the standardized precipitation index (SPI) based on precipitation, the evaporative demand drought index (EDDI) based on AED, and the standardized evaporation precipitation index (SPEI) based on the differences between precipitation and AED as flash drought indicators for mainland Spain and the Balearic Islands for 1961–2018. The results show large differences in the spatial and temporal patterns of flash droughts between indices. In general, there was a high degree of consistency between the flash drought patterns identified by the SPI and SPEI, with the exception of southern Spain in the summer. The EDDI showed notable spatial and temporal differences from the SPI in winter and summer, while it exhibited great coherence with the SPEI in summer. We also examined the sensitivity of the SPEI to AED in each month of the year to explain its contribution to the possible development of flash droughts. Our findings showed that precipitation is the main driver of flash droughts in Spain, although AED can play a key role in the development of these during periods of low precipitation, especially in the driest areas and in summer.

**Keywords:** flash drought; sensitivity; atmospheric evaporative demand; precipitation; standardized drought indices; SPEI; Spain





## 1. Introduction

Drought is one of the most complex natural hazards affecting natural and human systems [1–3]. Typically, drought is considered to be a meteorological phenomenon that is slow to develop, taking many months or even years to reach maximum intensity [4,5]. However, recent studies have proved that drought may also develop in the short-term as a result of large precipitation deficits and/or anomalous increases in atmospheric evaporative demand (AED) [6]. The term "flash drought" has become popular in the scientific literature to describe these drought episodes characterized by rapid onset and intensification. Flash droughts usually begin as a meteorological drought that becomes an agricultural drought within a short time, due to a decline in soil moisture and an increase in vegetation stress [7], with a strong potential impact on agriculture and the environment [8].

Precipitation is the most important factor in the development of droughts [9]. However, the occurrence of flash droughts has been commonly related to drastic increases in AED and the associated land–atmosphere feedbacks (e.g., increased land evapotranspiration) that can trigger or reinforce drought conditions [10]. Numerous studies have evidenced that AED can play a key role in the development and intensification of certain drought episodes [11–13]. Therefore, the AED can be a crucial variable in explaining the rapid depletion of soil moisture and vegetation stress caused by flash droughts [12,14–16].

However, given complex land–atmosphere feedbacks involving drought [10], the role of AED in the development of flash droughts could be very complex and variable in time and space [17].

Efforts to develop objective methodologies and early warning systems to identify flash droughts have increased in recent years [7,14–16,18–26], but none is widely acceptable [6]. Most methods proposed for identifying flash droughts mainly focus on rapid changes in reference evapotranspiration (ET$_o$) or soil moisture [7,19,23,24], without including precipitation as an input variable. Other authors, such as Mo and Lettenmaier [16,22] or Zhang et al. [27] suggested differentiating two types of flash droughts: "heat wave flash droughts" linked to high temperatures and "precipitation deficit flash droughts" linked to anomalously low rainfall.

In this work, we studied the role of precipitation deficit and atmospheric evaporative demand increase on flash droughts in Spain, where flash droughts are a frequent phenomenon with approximately four in every 10 drought events (i.e., a return period of 1 in 10 years) resulting in a flash drought [18]. Several studies showed the high occurrence of severe droughts episodes in Spain through historical [28–30] and instrumental records [31]. Spain has strong climatic contrasts with marked spatial and seasonal differences in precipitation [32–35] and AED [36–39]. Northern Spain is characterized by its humid oceanic climate with abundant precipitation over the year, while northeastern and southeastern regions are mainly characterized by semiarid conditions with annual precipitation generally below 300 mm [40]. On the other hand, the complex topography given by the presence of numerous mountain chains results in a strong continental features in central Spain. This climatic complexity is also reflected in the spatial and temporal variation of droughts [41,42] and the variety of atmospheric mechanisms triggering them [43–45], which enable an evaluation of the role of precipitation deficits and AED in the development of flash droughts over a wide range of climate conditions.

For this purpose, we compared three standardized drought indices based on different climate variables for flash drought analysis. The objectives of this study were twofold: (i) to compare the spatial and temporal patterns of flash droughts identified by three robust standardized drought indices based on different climatic variables: the standardized precipitation index (SPI), evaporative demand drought index (EDDI), and standardized evaporation precipitation index (SPEI); and (ii) to analyze the role of precipitation deficits and AED excess in flash droughts in Spain over the period 1961–2018.

## 2. Methods

### 2.1. Data

This research used a high spatial resolution (1.21 km$^2$) gridded climate dataset with coverage for mainland Spain and the Balearic Islands from 1961 to 2018 at weekly frequency. The climate dataset included data on precipitation, maximum and minimum air temperature, relative humidity, sunshine duration, and wind speed. The gridded dataset was created based on all daily observational information from the National Spanish Meteorological Service (AEMET) by means of an interpolation scheme of universal kriging using as input the meteorological data measured in the different meteorological stations and the terrain elevation. The climate series were subjected to a homogenization process and a careful quality control (see details in Tomas-Burguera et al. [46]). Additional information about the dataset development, interpolation methodology, and validation are available in Vicente-Serrano et al. [47]. The reference evapotranspiration (ET$_o$), as a metric of the AED, was calculated from the maximum and minimum temperature, relative humidity, wind speed, and sunshine duration, using the FAO-56 Penman–Monteith equation [48].

### 2.2. Computing the Drought Indices: SPI, EDDI, and SPEI

Standardized drought indices are commonly used in drought analysis and monitoring across the world. Among these, some of the most widely used for drought analysis are the standardized precipitation index (SPI; see Mckee et al. [49]) based on precipitation and the

standardized evaporation precipitation index (SPEI; see Vicente-Serrano et al. [50]) based on the difference between precipitation and AED. In particular, the SPI and SPEI also proved to be robust metrics for identifying flash droughts [51]. Recently, other standardized drought indices, such as the evaporative demand drought index (EDDI; see Hobbins et al. [52]) based exclusively on AED, was developed and recommended for flash drought analysis [19]. The EDDI quantify drought severity as AED increases under water-limited conditions, which is very useful during periods of low precipitation or soil moisture and important land–atmosphere coupling [17].

The SPI and SPEI were calculated using parametric approaches, fitting the data to Gamma and Log-logistic distributions, respectively. Moreover, the EDDI was calculated by a parametric approach based on Log-logistic distribution [53]. The SPI, EDDI, and SPEI are comparable in time and space and can be calculated on different time scales to adapt the response of hydrological, agricultural, and environmental systems to the climate variability [54]. However, the shorter scales better capture the rapid variations in precipitation and/or atmospheric evaporative demand that can trigger a flash drought, since the anomalies accumulated over past climate conditions do not affect the index values. Therefore, we calculated the SPI, EDDI, and SPEI at a short time scale (1-month) and a weekly frequency for each grid point of the climate dataset from 1961 to 2018 (see details in Noguera et al. [18]).

### 2.3. Identifying Flash Droughts Based on Standardized Drought Indices: SPI, EDDI, and SPEI

Flash drought events were identified following the methodology proposed by Noguera et al. [18]. The original approach was based on the SPEI at a short time scale (1-month) and high-frequency data (weekly) to identify the onset of flash drought episodes. This method focuses on the rapid development characteristic of flash droughts, which results in a sudden, very sharp drop in the index values. Thus, flash drought is defined as: (i) a minimum length of four weeks in the development phase; (ii) an $\Delta$SPEI (in 4 weeks) equal to or less than $-2$ z-units; and (iii) a final SPEI value equal to or less than $-1.28$ z-units (i.e., a 10-year return period). This method was used here to identify flash droughts based on the three standardized drought indices: SPI, EDDI, and SPEI.

### 2.4. Comparison of the SPI, EDDI, and SPEI

We analyzed the spatial and temporal variability of the SPI, EDDI, and SPEI by comparing the spatial and temporal behavior of the values of each index in each month of the year. Pearson's correlation coefficient (95% confidence level) was used to examine the relationship between SPI, EDDI, and SPEI series over time and space. Given that each weekly data is calculated based on the precipitation, AED, or climate balance ($D = P - AED$) data accumulated over four weeks (corresponding to a 1-month time scale), we used the weekly data for the last week of each month to calculate the correlation between the indices in order to reflect the variability over the whole month. The relationship between the frequency of annual and seasonal flash drought series recorded by the SPI, EDDI, and SPEI for each grid point from 1961 to 2018 was also examined by means of Pearson's correlation coefficient (95% confidence level). In addition, we analyzed the relationship between the total frequency of flash droughts identified by each index and the associated significance. The significance of Pearson's r coefficients was estimated using a Monte Carlo approach, in which the total number of flash droughts recorded by SPI, EDDI, and SPEI was correlated in 1000 random samples of 30 points from the entire dataset at annual and seasonal scales. We also examined the trend of annual and seasonal flash drought series recorded by the non-parametric Mann–Kendall statistic. Autocorrelation was included in the trend analysis by the modified Mann–Kendall trend test, which returned corrected *p*-values after accounting for temporal pseudoreplication [55,56]. To assess the magnitude of change in the frequencies of flash droughts, we used a linear regression analysis between the series of time (independent variable) and the seasonal and annual series of drought frequencies (dependent variable). The slope of the regression indicated the amount of

change in the number of events per year, with higher slope values indicating greater variation. In order to identify changes between the frequencies of flash droughts recorded for each index, we also calculated the differences (events/for each grid point) between the flash drought series obtained through the SPI, EDDI, and SPEI annually and seasonally and also examined their trends using the non-parametric Mann–Kendall statistic.

### 2.5. Evaluation of Sensitivity of Flash Droughts to AED

To analyze the role of precipitation deficits and atmospheric evaporative demand (AED) positive anomalies in the development of flash drought events in Spain, we calculated the sensitivity of SPEI to the AED at a short time scale (1-month) and compared it with the total frequency of flash drought events recorded in each month of the year in the period from 1961 to 2018. The sensitivity of SPEI to AED differs between climate conditions [9,17]. The SPEI is based on standardization of the climate balance (*D*) resulting from differences between precipitation and AED ($D = P - AED$), making it possible to quantify the contribution of precipitation and AED to the variability of SPEI values, following the methodology proposed by Tomas-Burguera et al. [9]. Thus, using the precipitation and AED series employed to compute SPEI, we calculated the partial derivatives of the climate balance (*D*) to determinate the relative contribution of both variables in each month over the period 1961–2018. The series of precipitation and AED were detrended prior to making the analysis to avoid the possible effects of trends on the results (see more details in Tomas-Burguera et al. [9]).

### 3. Results

### 3.1. Spatial and Temporal Variability of the SPI, EDDI, and SPEI

In general, the indices present high interannual variability in all months (Figure 1). The SPI and EDDI show non-significant correlation from November to January. On the contrary, there is a significant correlation between the SPI and EDDI from March to June and September–October (Pearson's r > 0.7). February, July, and August returned lower correlation values, although there was a significant correlation. Correlation between the EDDI and SPEI was also low from November to February, although with slightly higher correlation values than those found between the SPI and EDDI. From March to October, there was a high correlation between EDDI and SPEI, reaching a maximum in May, June, and July (Pearson's r ≥ 0.9). Correlation between the SPI and SPEI was very high and significant in all months of the year, although it was slightly lower in July and August.

The spatial pattern of the correlation and associated significance between monthly series of the SPI, EDDI, and SPEI over the period 1961–2018 also show some relevant differences (Figure 2). In general, there is a low, non-significant correlation between the SPI and EDDI in northwestern regions from November to January, while it is high and significant in the Mediterranean coastland and southern Spain. From February to June, also in September and October, there is a high, significant correlation between SPI and EDDI. In contrast, correlation between the SPI and EDDI was low and non-significant in July and August in large areas of southern Spain. Correlation between the EDDI and SPEI was also low and non-significant in northwestern regions, but it was high and significant in southern and eastern Spain from November to January. From February to October, there was a significant correlation between the EDDI and SPEI over most of the study area, reaching a maximum from May to July. As expected, there was very high and significant correlation between the SPI and SPEI in every month over almost all of the study area, with only some areas in southern Spain, where precipitation is very low in summer, returning low values in July and August.

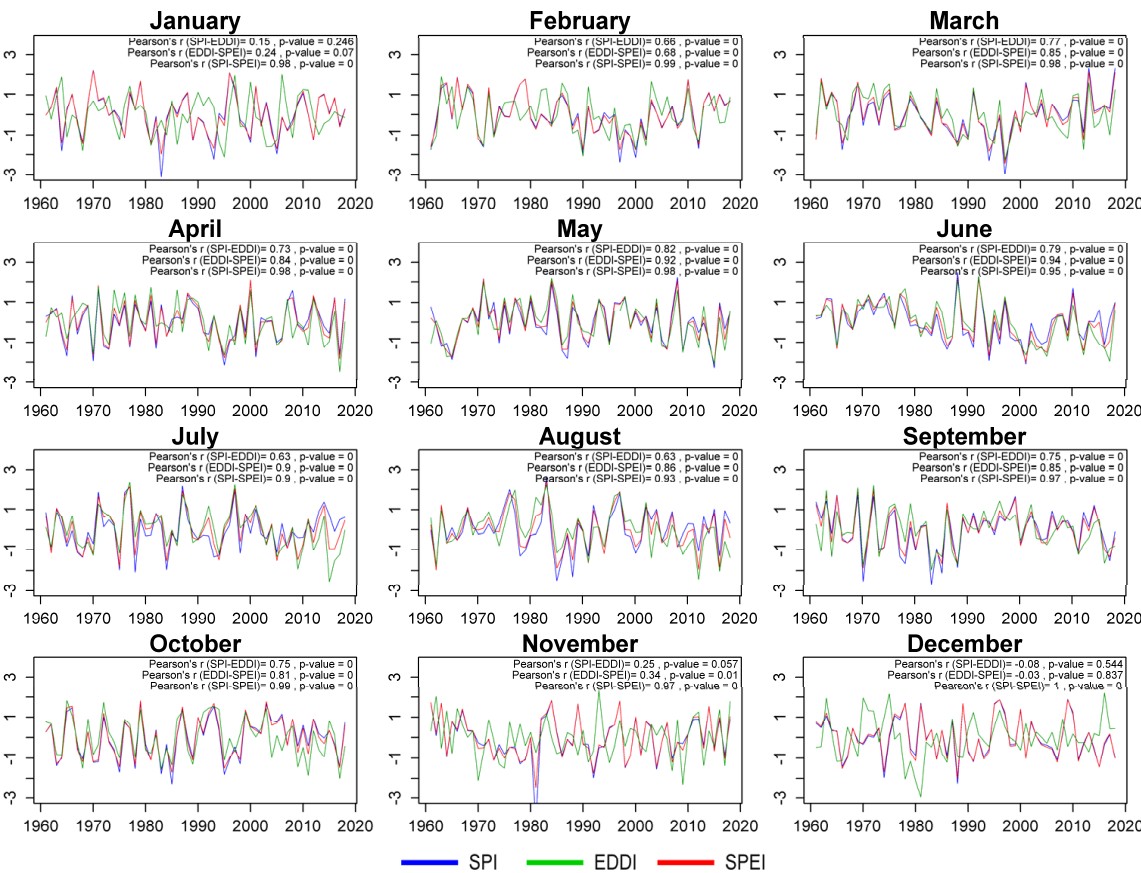

**Figure 1.** Temporal evolution of monthly series of the standardized precipitation index (SPI), evaporative demand drought index (EDDI), and standardized evaporation precipitation index (SPEI) based on average data on mainland Spain and the Balearic Islands over the period 1961–2018 at a short time scale (1-month). The monthly series includes the weekly data for the last week of each month in each year.

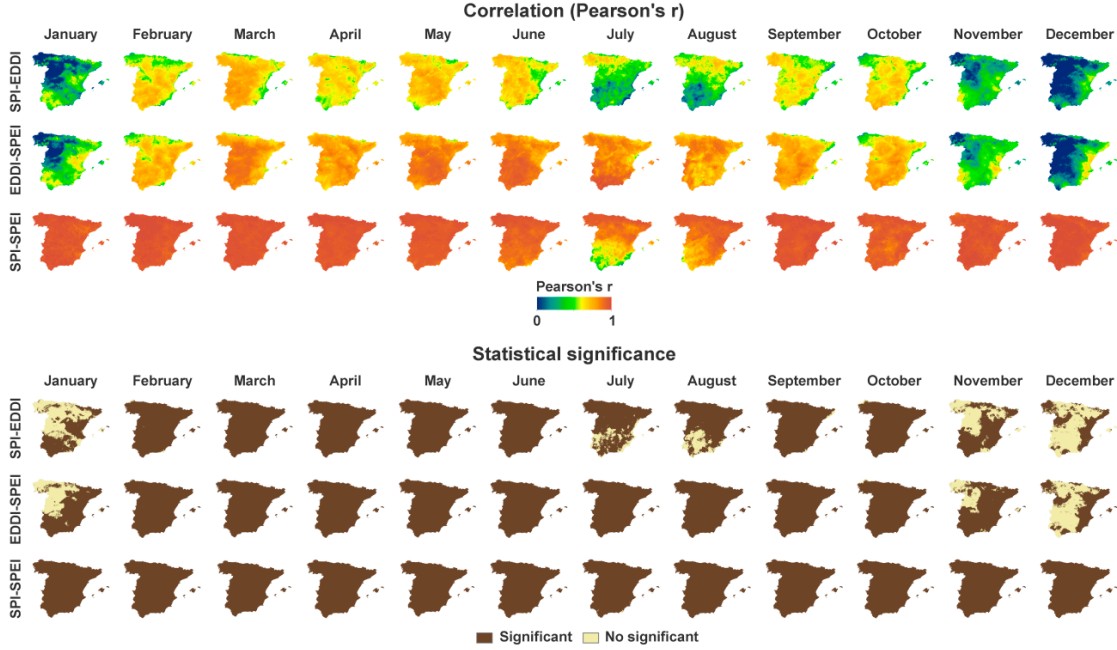

**Figure 2.** Spatial pattern of the correlation (Pearson's r) and associated significance between monthly series of the SPI, EDDI, and SPEI over the period 1961–2018. The monthly series include the weekly data for the last week of each month in each year.

*3.2. Spatial Distribution and Trends in Flash Drought*

Figure 3 shows the spatial distribution and frequency of flash droughts (events/for each grid point) identified by the SPI, EDDI, and SPEI from 1961 to 2018 at annual and seasonal scales. There were notable differences in the spatial distribution of flash droughts identified by the SPI, EDDI, and SPEI (Figure 3a), as well as in the average frequency of events recorded for each index (Figure 3b). The average frequency of flash droughts obtained by the SPI ($\approx$70 events/for each grid point) was considerably higher than for the EDDI and SPEI ($\approx$40 events/for each grid point) at an annual scale. In general, the SPI identified a high occurrence of flash droughts in most of the study area at an annual scale, which was also noted in some areas of central and northeastern Spain, while the Mediterranean coastland recorded the lowest number of events. The EDDI and SPEI recorded the highest number of flash drought events in northern and northwestern Spain, although the EDDI also recorded a great many in some areas of the Mediterranean coastland.

At a seasonal scale, there were also spatial differences between patterns of the average frequency of flash droughts recorded by the SPI, EDDI, and SPEI. In winter, similar spatial patterns were found for SPI and SPEI, with the highest occurrence found in the northern and eastern regions of the Iberian Peninsula and in the Balearic Islands. However, the average frequency of flash droughts identified by the SPI ($\approx$16 events/for each grid point) was substantially higher than by the SPEI, which recorded the lowest average frequency in this season ($\approx$9 events/for each grid point). On the other hand, the EDDI recorded a great many flash droughts in large areas of central Spain in winter, reaching its maximum average frequency for the season, with approximately 13 events/for each grid point. In spring, the spatial distribution and average frequency of flash droughts recorded by the SPI showed great disparities with the EDDI and SPEI. The average frequency of the flash droughts recorded by the SPI ($\approx$17 events/for each grid point) was considerably higher than by the EDDI and SPEI ($\approx$10 events/for each grid point). The SPI recorded a high incidence of flash droughts in spring in most of central and western Spain, and also in some areas of the Pyrenees. In contrast, the EDDI recorded a low number of flash droughts in those regions, and more events were found in certain areas of the Mediterranean coastland and northern and northwestern Spain. Similarly, the SPEI identified a high occurrence of flash drought events in northern and northwestern regions but also in large areas of southern Spain. In summer, the SPI and EDDI recorded their lowest average frequency, while the SPEI reached its maximum average frequency ($\approx$12 events/for each grid point). There is no clear spatial pattern of flash drought identified by the SPI in summer, with a high number of flash droughts occurring in central, northwestern, and northeastern Spain. The EDDI identified the highest number of flash droughts in northern and western regions, while central and eastern Spain had few events. In general, the SPEI recorded a high frequency of flash drought events in summer, with wide areas of the south and northwest exceeding 15 events/for each grid point. In autumn, the pattern of spatial distribution of flash droughts identified by the SPI and SPEI was similar in northern Spain, and both recorded a low number of events in the Mediterranean coastland. However, the SPI also identified a high number of events in large areas of central Spain, reaching its maximum average frequency in this season with approximately 19 events/for each grid point. Unlike the SPI and SPEI, the EDDI recorded the highest incidence of flash droughts in the Mediterranean coastland in autumn, but it only recorded a few in western Spain.

In general, there was non-significant spatial correlation between the total number of events identified for each index and the shared variance was not high, due to notable spatial differences annually and seasonally (Figure 4). At an annual scale, there was a negative and non-significant correlation between the spatial distribution of the total number of flash droughts recorded by the SPI and EDDI. The highest correlation was found between the EDDI and SPEI at annual scale, although it was non-significant. The SPI and SPEI also returned a positive, but non-significant, correlation annually.

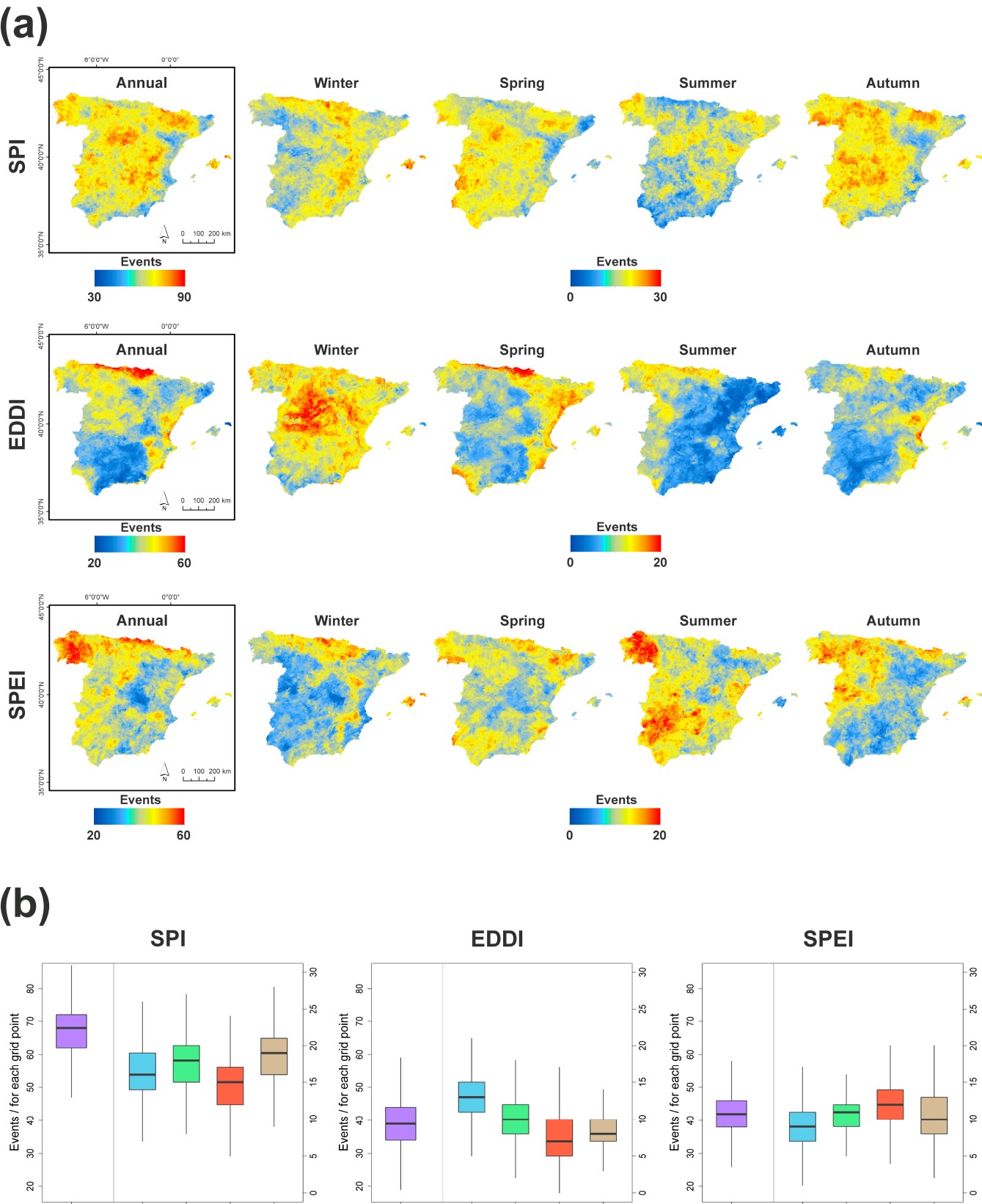

**Figure 3.** Annual and seasonal (**a**) spatial distribution for the frequency of flash droughts (events/for each grid point) identified by the SPI, EDDI, and SPEI over the period 1961–2018, (**b**) box plots summarizing the annual and seasonal frequencies obtained with the different indices. In the box plot, the left axis represents the annual data and the right axis represents the seasonal data.

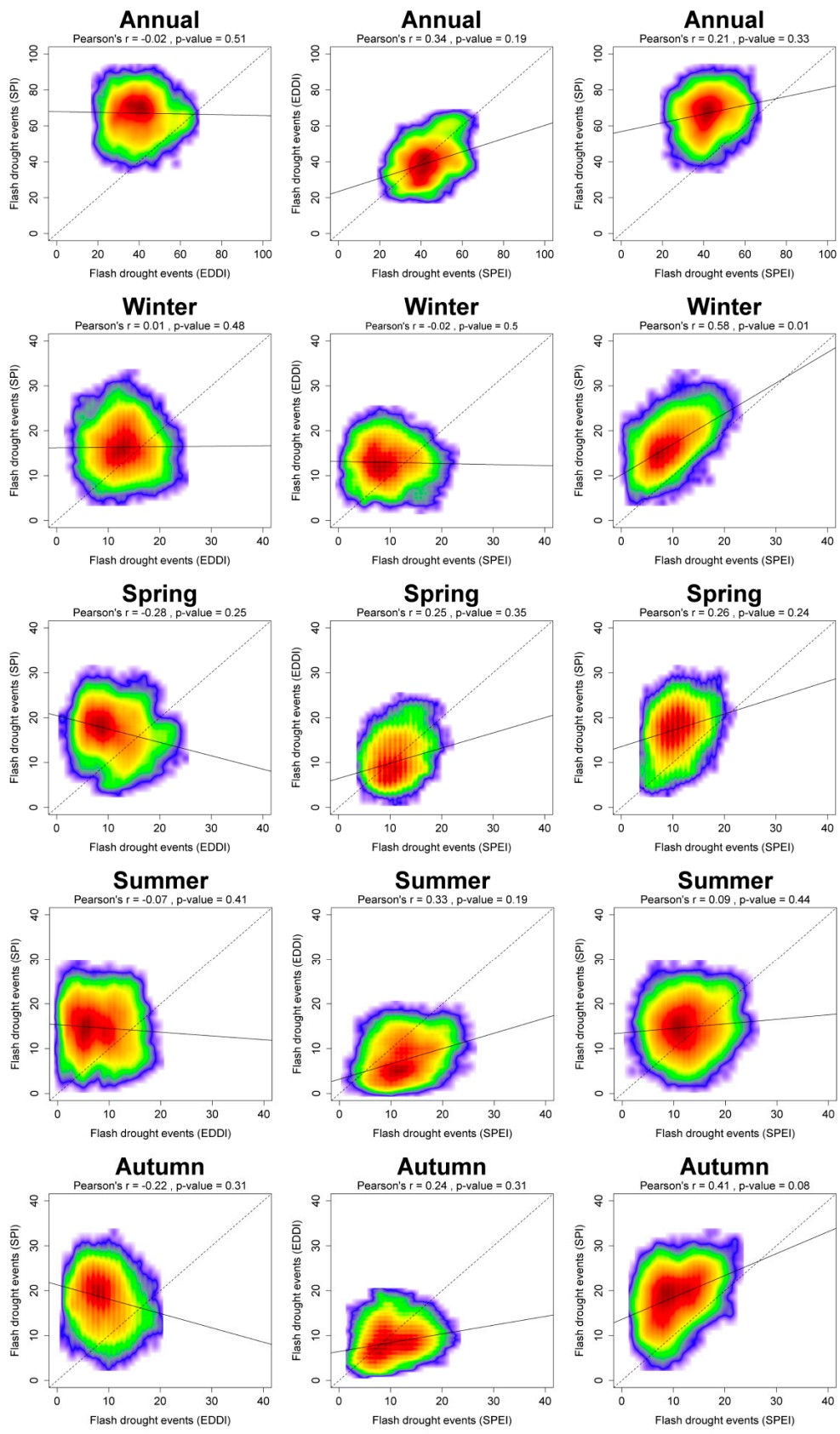

**Figure 4.** Scatterplots showing the annual and seasonal relationship between the total number of flash drought events recorded by the SPI, EDDI, and SPEI. The colors represent the density of points, with red denoting the highest density.

At a seasonal scale, there was a negative and non-significant correlation between the spatial distribution of the total number of flash droughts recorded by the SPI and the EDDI in all seasons, except in winter. Correlation between the EDDI and SPEI in winter was negative and non-significant, while it was positive and non-significant in spring, summer, and autumn. The highest correlation between the EDDI and SPEI was found in summer (Pearson's r = 0.33). The spatial distribution of the total number of flash droughts recorded by the SPI and SPEI showed a positive correlation in all seasons, reaching the maximum values in winter (Pearson's r = 0.58) and autumn (Pearson's r = 0.41), although it was only significant in winter.

Figure 5 depicts the spatial distribution of the magnitude of change and significance of trends in annual and seasonal frequency of flash droughts recorded by the SPI, EDDI, and SPEI over the period 1961–2018. At an annual scale, there was a significant decline in the number of flash droughts identified by the SPI across wide areas of central Spain, and only small areas of the Mediterranean coast and northern Spain showed significant increases in flash droughts. The EDDI also recorded a decrease in flash droughts in some areas of central Spain, but this decline was generally non-significant. In contrast, there was a significant increase in the frequency of flash drought events identified by the EDDI in some areas of southern and northwestern Spain. The SPEI also identified significant increases in flash droughts in southern Spain annually, as well as in some areas of the Mediterranean coastland. Similar to the SPI and EDDI, the SPEI also recorded a general decrease in flash droughts in central and northern Spain, although this was only statistically significant in a few areas.

At a seasonal scale, most of the study area showed non-significant trends in the occurrence of flash droughts recorded by the SPI, EDDI, and SPEI over the period 1961–2018. The three indices identified a general decline in flash drought events in central Spain in winter, although only the SPI and EDDI found statistically significant decreases in some of these areas. In spring, similar trends were found by the SPI, EDDI, and SPEI, with negative and non-significant trends in most of the study area. Furthermore, the three indices recorded increases in the frequency of flash droughts in the Mediterranean coastland and Balearic Islands in this season, but only the SPEI and the SPI showed positive and statistically significant trends. In summer, there were notable differences in flash drought trends recorded by the SPI, EDDI, and SPEI. The SPI identified negative, non-significant trends in most of the study area, while the EDDI and SPEI showed a general increase in flash droughts. The SPEI identified some significant increases in flash droughts in southern regions in summer, while the EDDI recorded statistically significant increases in a few small areas of western Spain. In autumn, the SPI returned a general decrease in flash droughts in most of the study area, although this was only significant in northeastern Spain. On the contrary, the EDDI and SPEI identified positive, non-significant trends in the occurrence of flash drought in most of the study area, with the exception of northern regions, where negative, non-significant trends were noted.

The temporal evolution of the average frequency of flash drought events identified by the SPI, EDDI, and SPEI on mainland Spain and the Balearic Islands over the period 1961–2018 showed a high variability at annual and seasonal scales (Figure 6). The annual series of flash droughts recorded by the SPI, EDDI, and SPEI exhibited statistically significant correlations among them, although the average frequency of flash droughts identified by the SPI was substantially higher until the year 2000. There was a significant decline in flash droughts recorded by the SPI annually, while the annual series obtained by the SPEI and EDDI showed non-significant trends.

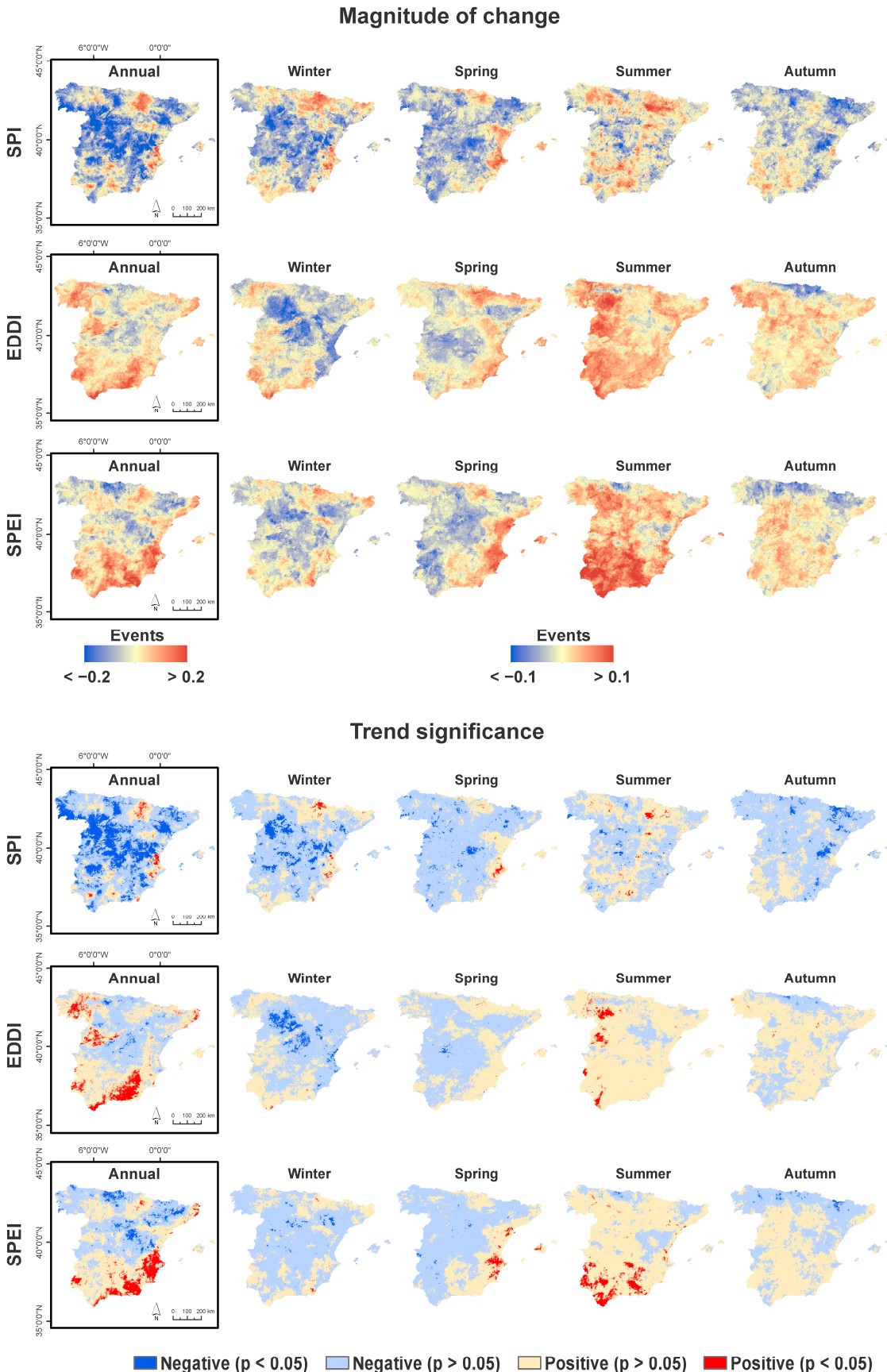

**Figure 5.** Spatial distribution of the annual and seasonal magnitudes of change per decade and the significance of trends of flash drought events identified by the SPI, EDDI, and SPEI over the period 1961–2018.

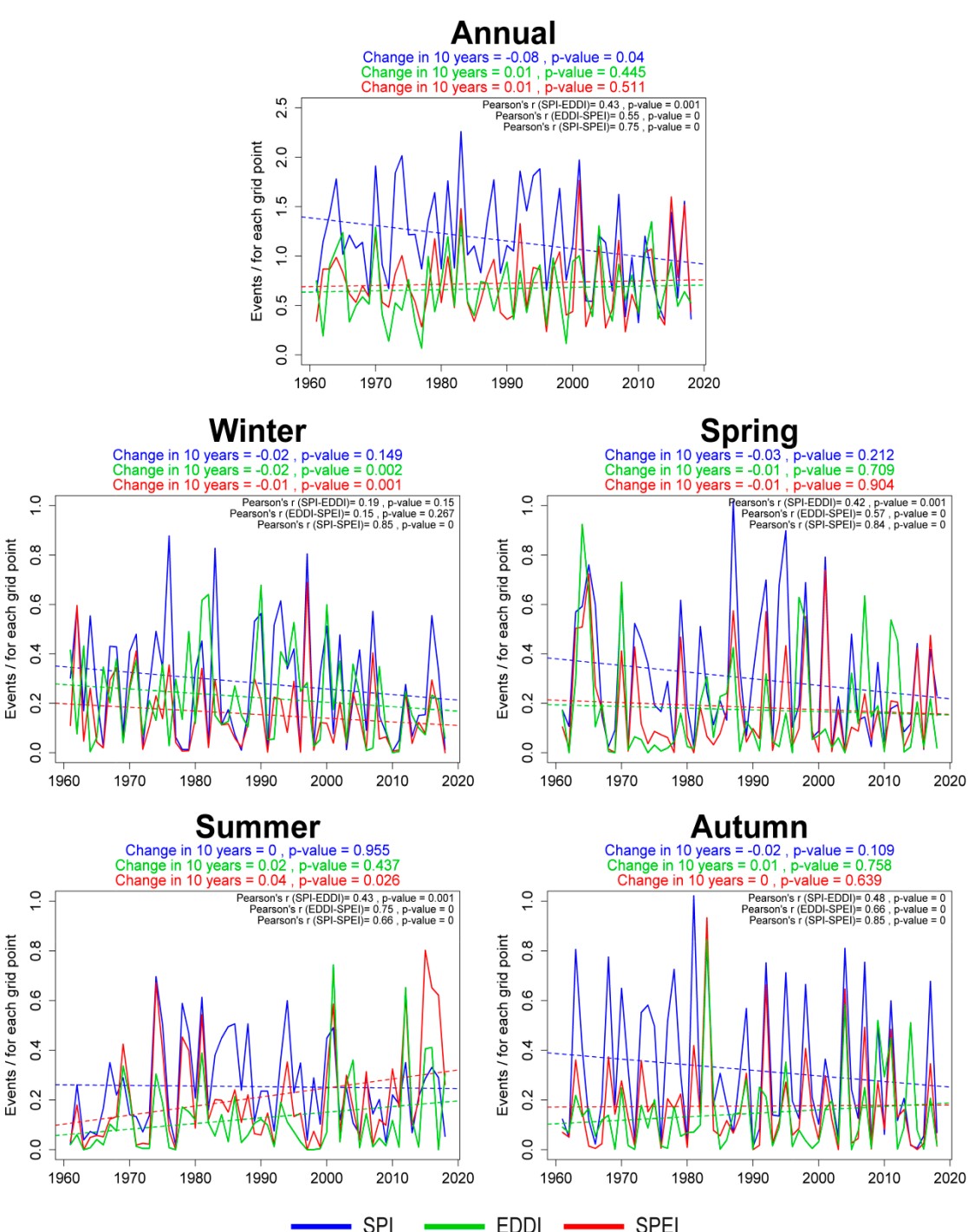

**Figure 6.** Temporal evolution of the average frequency of flash droughts identified by the SPI, SPEI, and EDDI on mainland Spain and the Balearic Islands over the period 1961–2018 at an annual and seasonal scale.

At a seasonal scale, the flash drought series recorded by the SPI, EDDI, and SPEI from 1961 to 2018 also exhibited high interannual variability. In winter, the flash drought series returned by the three indices showed negative trends, although this was only statistically significant for the EDDI and SPEI. The flash drought series recorded by the EDDI showed a low, non-significant correlation with the SPI and SPEI in this season, while there was a significant correlation between the SPI and SPEI (Pearson's r = 0.85). In spring, a negative,

non-significant trend was noted in the series from the SPI, EDDI, and SPEI. The flash drought series also showed significant correlation in this season; the highest correlation was found between the SPI and SPEI (Pearson's r = 0.84). In summer, there was a positive trend in flash droughts recorded by the EDDI and SPEI, but it was only statistically significant for the SPEI. The flash droughts series obtained with the SPI showed a non-significant trend. Summer also returned a significant correlation between the flash drought series recorded by the SPI, EDDI, and SPEI, although in this case, higher correlation was noted between the EDDI and SPEI. In autumn, the series recorded by the SPI, EDDI, and SPEI showed non-significant trends. There was a significant correlation between the series obtained by the three indices, with the highest found between the SPI and SPEI (Pearson's r = 0.85).

The difference in the number of flash droughts recorded between the SPI and SPEI, and also between the SPI and EDDI, showed a significant decrease over the period 1961–2018 at an annual scale (Supplementary Materials Figure S1). This is a result of the decline in the number of flash droughts recorded by the SPI over the last two decades, which explains why the average frequency events/for each grid point reported by the three indices was very similar in recent years. On the other hand, the differences between the series of flash drought frequency recorded by the EDDI and SPEI annually did not show a significant trend over the study period.

At a seasonal scale, the differences among series obtained by the SPI, EDDI, and SPEI also exhibited some notable changes over the period 1961–2018 (Supplementary Materials Figure S1). For example, spring exhibited a significant decrease in the difference in events identified by the SPI and the SPEI. In summer, non-significant trends were found, although there was a decrease in the difference in events identified by the SPI and SPEI as a result of the increase in flash drought events recorded by the SPEI over the last few years. In autumn, the difference between the flash drought series identified by the SPI and EDDI, and also between the SPI and SPEI, showed a negative trend, although it was only statistically significant between the SPI and the SPEI. On the other hand, there was a non-significant trend in the difference in the number of flash drought events identified by the EDDI and SPEI in winter and autumn.

Figure 7 presents the spatial pattern of the correlation and associated significance between flash drought series recorded by the SPI, EDDI, and SPEI over the period 1961–2018 at annual and seasonal scales. In general, the annual flash drought series obtained by the SPI and EDDI showed a low, non-significant correlation in most of the study area. In contrast, there was a significant correlation between the series recorded by the EDDI and the SPEI at an annual scale, especially in areas of southern and northeastern Spain, where higher correlations were found. The annual flash drought series from the SPI and SPEI returned a high and significant correlation in most of the study area, although a stronger spatial correlation was found in northern regions (Pearson's r > 0.8).

At a seasonal scale, the spatial correlation between flash drought series recorded by the SPI, EDDI, and SPEI showed notable differences. In winter, a low, non-significant correlation was found between the series recorded by the SPI and EDDI. This was the same for the series obtained by the EDDI and SPEI. On the contrary, the SPI and SPEI showed a very high and significant correlation in most of the study area in winter, with the exception of some areas of the Mediterranean coastland and the Ebro Depression. In spring, the correlation between flash drought series from the SPI and EDDI was generally low and non-significant, although in some areas of northwestern, Spain it was high and significant. In contrast, there was a high and significant correlation between flash drought series recorded by the EDDI and SPEI, especially in the Mediterranean coastland. The series recorded by the SPI and SPEI also indicated a high and significant correlation in most of the study area in spring. In summer, the correlation between series obtained through the SPI and EDDI was generally low and non-significant. However, those series recorded by the EDDI and SPEI indicated a very high and significant correlation in this season, especially in southern regions and areas of northwest and north of Spain. The flash drought series recorded by the SPI and SPEI generally exhibited a high and significant correlation

in central and northern regions in summer, while these were low and non-significant in southern Spain. In autumn, the correlation between the flash drought series recorded by the SPI and EDDI was low and non-significant, with the exception of some areas of central Spain. The series obtained from the EDDI and SPEI displayed a high correlation across large areas in the study, although these were low and non-significant in the northern regions, and also in some areas of southern and central Spain. On the other hand, the flash drought series identified by the SPI and SPEI showed a high and significant correlation in autumn, with the exception of few areas of southeastern Spain.

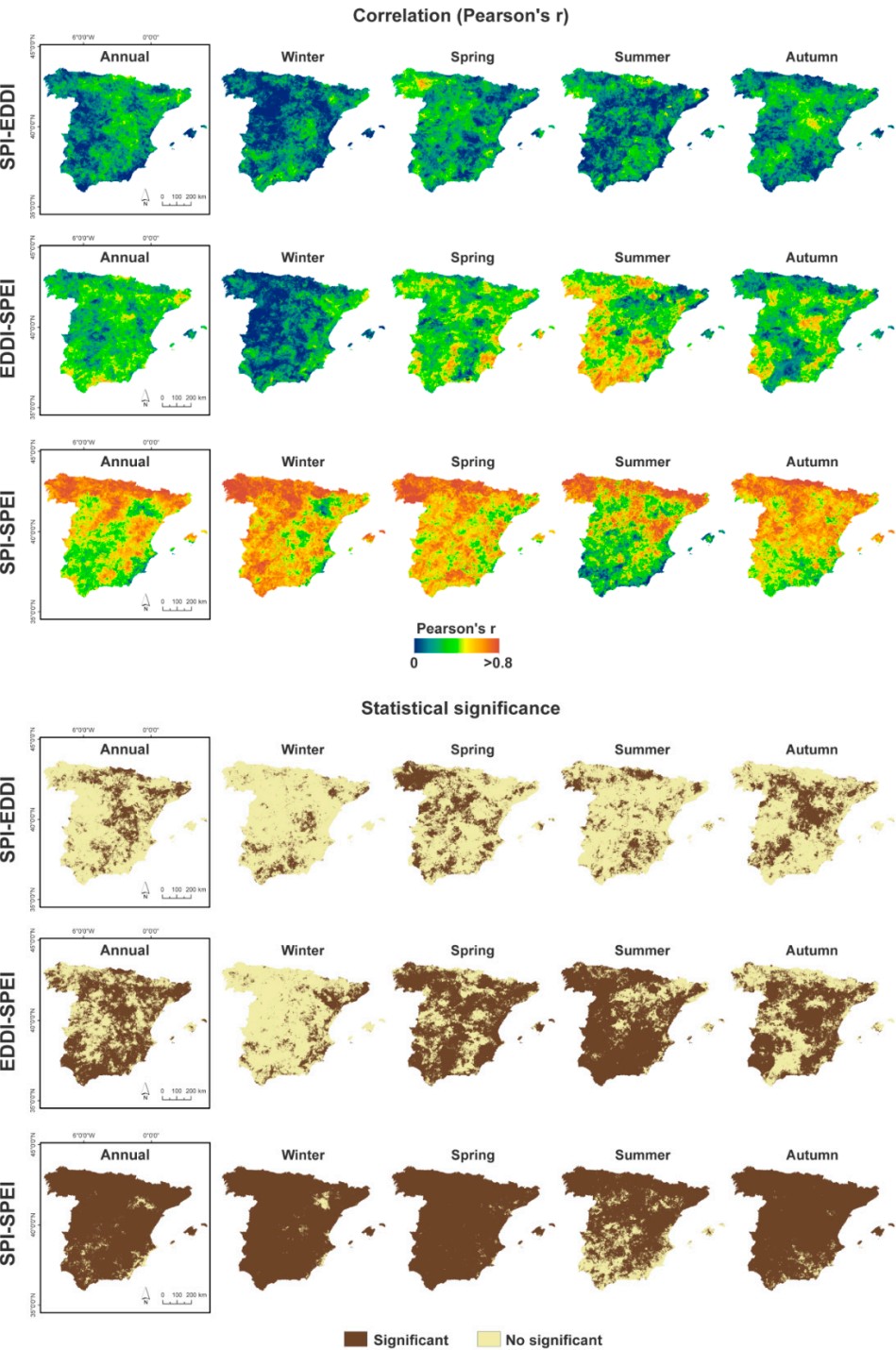

**Figure 7.** Spatial pattern of the correlation (Pearson's r) and associated significance between flash drought series identified by the SPI, EDDI, and SPEI over the period 1961–2018 at annual and seasonal scale.

### 3.3. Sensitivity of Flash Droughts to AED

To determine the differential sensitivity of the flash droughts recorded in different periods of the year, we analyzed the sensitivity of the 1-month SPEI to changes in precipitation and the AED. Table 1 shows the sensitivity SPEI values to AED on mainland Spain and the Balearic Islands from 1961 to 2018 at a 1-month time scale. As expected, the sensitivity of SPEI to AED showed noticeable seasonal contrasts. SPEI values display low sensitivity to AED in winter (December, January, and February), and the values were close to zero in December and January. Spring indicated a moderate sensitivity of the SPEI to AED with values increasing from March (14.35%) to May (34.60%). The highest was noted in June, July, and August, reaching maximum values in July of 46.68%. In autumn, there was a gradual decrease from September (23.11%) to November (2.23%).

**Table 1.** Sensitivity of the SPEI to atmospheric evaporative demand (AED) on mainland Spain and the Balearic Islands over the period 1961–2018 at a short time scale (1-month). The monthly series include the weekly data for the last week of each month in each year.

| SPEI 1-month | AED Sensitivity (%) |
|:---:|:---:|
| January | 1.37 |
| February | 6.54 |
| March | 14.35 |
| April | 22.13 |
| May | 34.60 |
| June | 37.98 |
| July | 46.68 |
| August | 37.06 |
| September | 23.11 |
| October | 12.50 |
| November | 2.23 |
| December | 0.50 |

The monthly spatial distribution of the sensitivity of SPEI to AED from 1961 to 2018 at a 1-month time scale also presents notable contrasts (Figure 8). There is a gradual increase in the sensitivity of the SPEI to AED from winter to summer, when the highest values were observed, and this increase was followed by a subsequent gradual decrease from summer to winter (Supplementary Materials Figure S2). Similarly, there is a large spatial difference in the sensitivity of the SPEI to AED values during the warm season. In winter, when the AED in Spain is low, the SPEI showed the lowest sensitivity to AED, with values below 10% across most of the study area, meaning that flash droughts during these months are mostly determined by precipitation anomalies. In spring, the role of the AED increases, mostly in May, with average sensitivity values of around 30%. Therefore, the precipitation deficits play a principal role in the occurrence of flash droughts, although the slight increase in sensitivity of the SPEI to AED was noted during the late spring in southern Spain. In summer, there is higher sensitivity of the SPEI to AED in large areas of the study domain, with a marked south–north gradient. The maximum sensitivity of the SPEI to the AED was noted in July, with values above 70% in southern Spain. In autumn, the sensitivity of the SPEI to AED is generally low over most of the study area, with average values below 10% except in September (≈18%).

Figure 9 illustrates the relationship between the total number of flash drought events recorded using the SPEI and its sensitivity to AED in each month of the year over the period 1961–2018 at a short time scale (1-month). As indicated, from October to January, the sensitivity of the SPEI to AED is very low, even in those areas where most of the flash droughts were identified, so the role of precipitation in the development of flash droughts in those months seems clearly dominant. From February to April, the sensitivity of the SPEI to AED is slightly greater, and the areas where it was higher also recorded a greater occurrence of flash droughts, which clearly suggests that AED is highly relevant in the development of certain flash drought events during this time. Nonetheless, as the average

sensitivity values are generally below 20%, the dominant role of precipitation during spring seems clear, regardless of the number of flash droughts recorded. In contrast, from May to August, sensitivity to AED is notably stronger and wide areas with a high frequency of flash drought events also showed high sensitivity to AED, so the contribution of AED in the development of flash droughts recorded during these months is expected to be very important. However, there are also areas of northern Spain with low sensitivity that recorded a high incidence of flash drought events. This suggests large spatial differences in the drivers of flash droughts and also that the role of precipitation is crucial for the development of these in north Spain during summer.

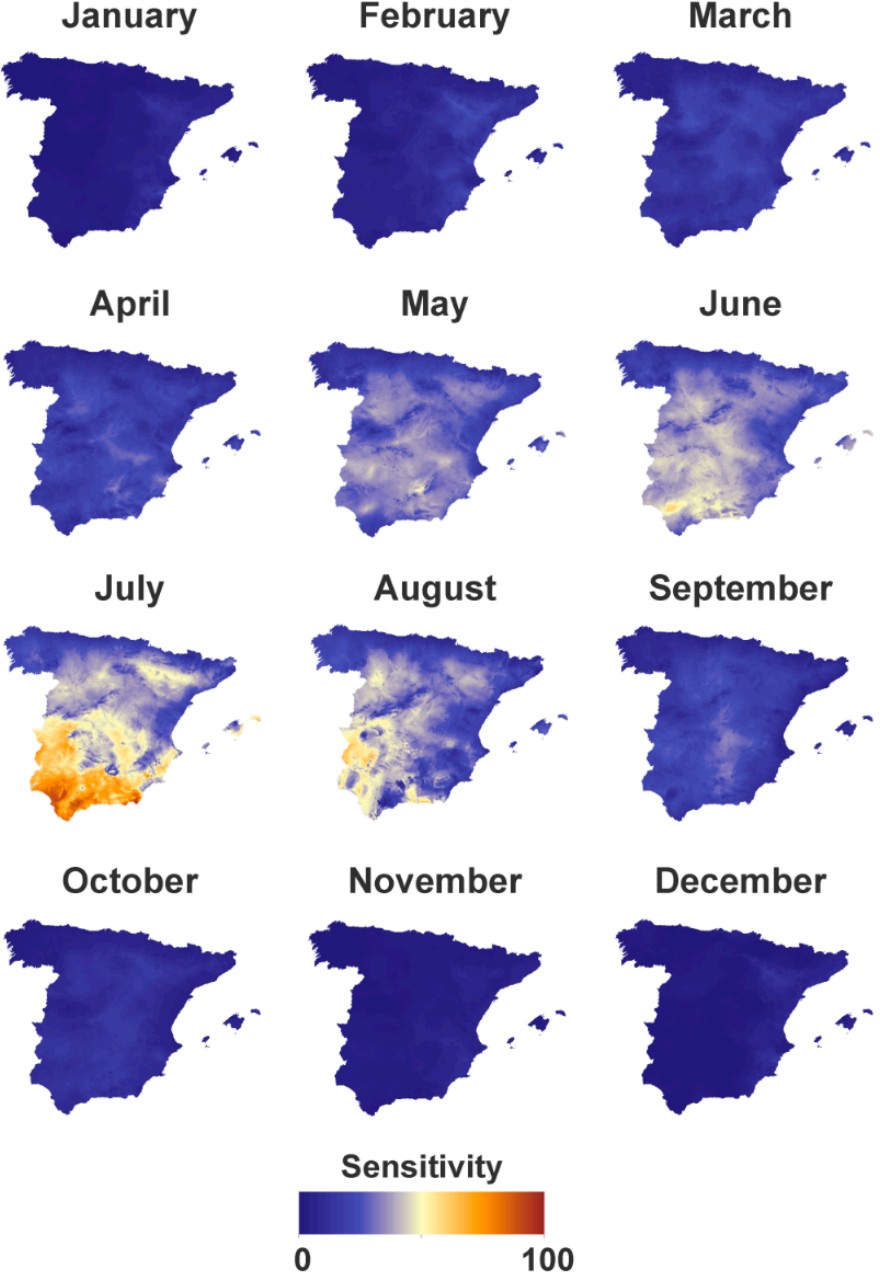

**Figure 8.** Monthly spatial distribution of the sensitivity (%) of the SPEI to AED on mainland Spain and the Balearic Islands over the period 1961–2018 at a short time scale (1-month). The monthly series include the weekly data for the last week of each month in each year.

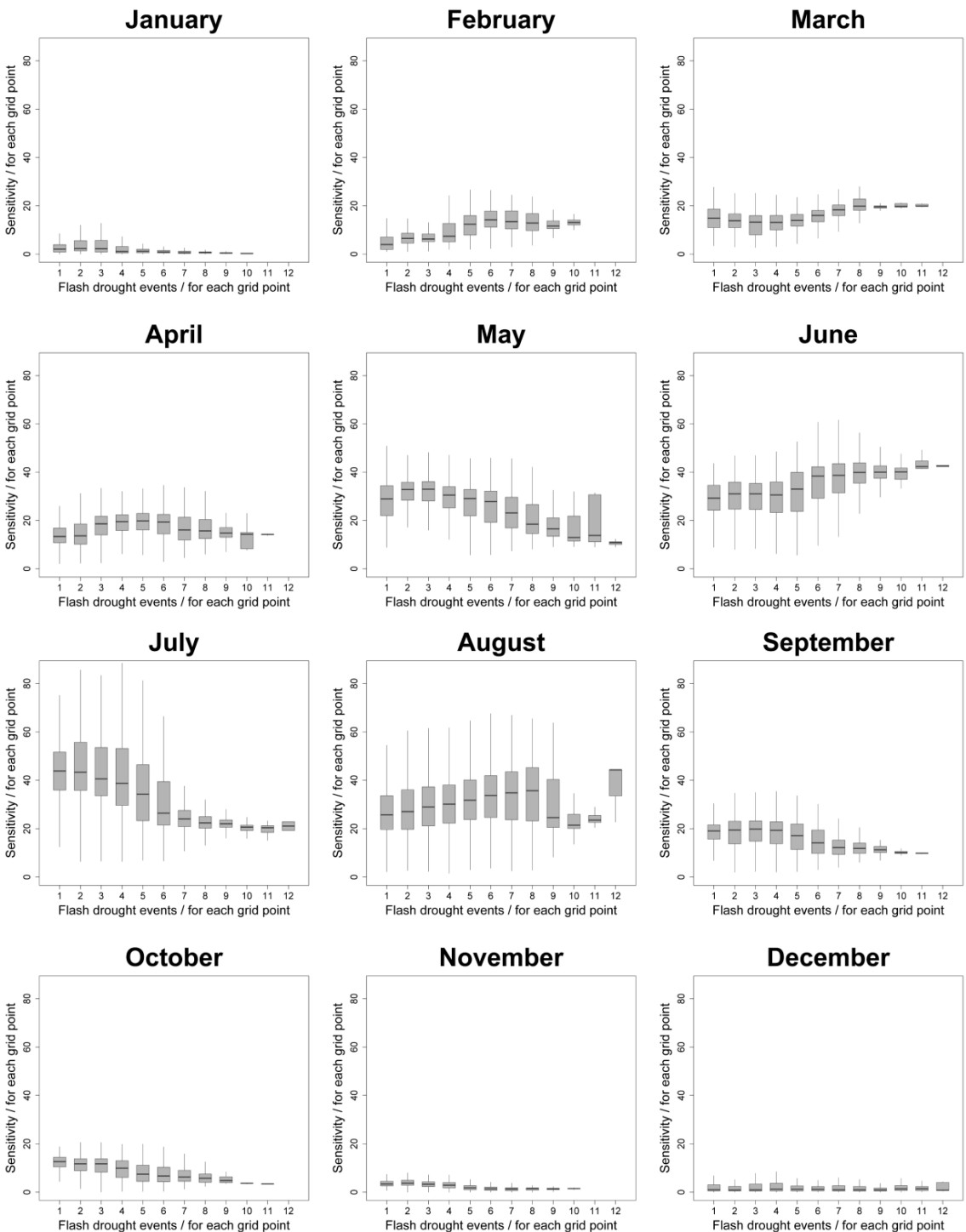

**Figure 9.** Monthly frequency of flash drought events/for each grid point obtained from the SPEI and its sensitivity (%) to AED on mainland Spain and the Balearic Islands in each month over the period 1961–2018 at a short time scale (1-month). Frequencies of flash drought events with a residual number of cases are not represented.

## 4. Discussion

In this study, we identified flash droughts based on different drought indices including the SPI, the EDDI, and the SPEI. This enabled the role of precipitation and AED in the development of flash droughts to be established, in addition to how this role determines the spatial and temporal patterns of flash droughts in Spain. The main advantage of these standardized indexes in comparison to other metrics is that they are comparable

in time and space, making it possible to apply the same methodology and to obtain comparable results based on indices from different variables. Furthermore, these indices also enable flash drought analyses in the long-term, since they are based on widely available climate information.

Numerous studies proved the usefulness of widely used, standardized drought indices such as the SPI or SPEI in identifying different types of impacts [57–60]. Previous studies also proved the robustness of standardized drought indices, such as the SPI, EDDI, and SPEI in identifying and characterizing flash droughts over different regions of the world [18,19,51,61]. Although it is not easy to determine which metric is more suitable for the analysis of flash droughts, we think that the implementation of different indices for flash drought analysis can be useful for a more comprehensive understanding.

The spatial and temporal behavior of flash droughts in Spain is highly complex and variable [18]. The results obtained in this research suggested that flash droughts in Spain can develop from both precipitation deficits and anomalous increases in AED, although there is a differentiated seasonal and spatial behavior. This is reflected in the notable differences found between the spatial and temporal patterns of flash droughts identified by the SPI, EDDI, and SPEI. Thus, even when the drought indices showed high spatial and temporal correlation in most months of the year, the spatial and temporal frequency of flash drought events was dependent on the index used, since although there is consistency among the three indices, there are large seasonal differences. For example, in winter, in which AED is low and drought conditions are closely related to precipitation deficits, correlation between the EDDI and SPI is low. Therefore, it is not recommended to use the EDDI to assess drought severity and, particularly, to identify flash droughts during winter. In autumn and spring, when the AED is slightly higher but precipitation is also the main driver controlling variations in index values, correlation of the EDDI with the SPI and SPEI was generally high, especially with the SPEI. This suggests an increase in the influence of the AED on drought severity. In summer, there is higher consistency between the EDDI and SPEI, with correlations similar to those found between the SPI and SPEI. This means that changes in AED can be as relevant or even more so than precipitation deficits in the response of indices during the summer, particularly in the driest areas. This indicates that in summer, the use of indices based exclusively on precipitation is not suitable to assess drought severity, and they are particularly unsuitable to identify flash droughts. The SPI showed a high spatial and temporal coherence with the SPEI in almost every month of the year, reflecting the fact that the SPEI responds mainly to variations in precipitation, except in the dry summer period. These findings prove the sensitivity of the SPEI to changes in AED during dry periods, and they are very consistent with the patterns in the response of the SPEI to AED shown by Tomas-Burguera et al. [9] at a global scale.

In general, there are noticeable spatial differences in the occurrence of flash droughts identified for each index. The spatial patterns recorded by the SPI and SPEI were strongly coherent in winter, particularly in northern Spain. This suggests that during wet periods, in which the role of precipitation deficits in drought development is not in doubt [17], precipitation is the main driver of flash droughts. However, the role of the atmospheric evaporative demand (AED) on droughts is more complex and exhibits large spatial and seasonal differences [17,62], making it difficult to determine its role in the development of flash droughts [63]. Several studies showed that flash droughts can be driven by anomalous increases in AED associated with heat waves and land–atmosphere feedbacks that cause or reinforce the rapid depletion of soil moisture [12,14–16]. Thus, the occurrence of flash drought is usually associated with strong anomalies of AED [6,19]. Here, we have shown that AED can play an important role in the warm season. In summer, when precipitation is generally very low in Spain [40,64], there were notable differences in the spatial patterns of flash droughts found by the SPI and SPEI. Compared to the SPI, the SPEI showed a greater spatial consistency with the EDDI, indicating that during the warm season, the AED is a key variable in explaining the occurrence of flash droughts. The physical processes explaining the importance of this variable can be diverse. Some studies showed that in periods of

very low soil moisture and strong land–atmosphere coupling, AED would be driven by the limited latent heat fluxes from a dry soil, which could favor some self-intensification of drought conditions [10,62]. However, in Spain, it is more probably due to the dominant role of warm, dry air advections originating in the Sahara, which are very frequent in summer [65].

In spring and autumn, opposite spatial patterns were found for the flash droughts identified by the SPI and EDDI. This could be related to a variable contribution of precipitation and AED in the development of the flash droughts, which seems reasonable considering the wide spatial and temporal variability of precipitation [32–35] and AED [36–38] during these seasons. Thus, it is possible that AED plays an important role in the occurrence of flash droughts in the Mediterranean coastland and northeastern Spain where the EDDI recorded a high incidence in spring and, to a lesser extent, in autumn. However, it would be secondary to precipitation in most of Spain. This is also reflected in the spatial patterns of the number of flash droughts identified by the SPEI, since there is spatial consistency with the SPI in most of the study area in spring and especially in autumn, but the SPEI also reported a high frequency of flash droughts in the Mediterranean coastland in spring, which is a pattern also found in the EDDI. Although in some cases, the different indices showed similar spatial patterns at an annual and seasonal scale, the shared variance was not generally high. This seems to confirm that the flash droughts in Spain can be triggered by different drivers, as well as the role of precipitation and AED being seasonally and spatially variable.

Only the SPI reported a negative and significant trend in the frequency of flash droughts, whereas the EDDI and SPEI recorded a non-significant trend for these events from 1961 to 2018. However, the EDDI and SPEI reported significant increases in the frequency of flash droughts in some areas of southern and southeastern Spain. Since the SPI reported non-significant trends in summer over most of the study area, these increases must necessarily be related to an increase in the contribution of the AED to the development of flash droughts in these areas. This hypothesis seems coherent with various studies that recorded increased AED in Spain during the summer [36,38]. Some areas of the Mediterranean coastland also showed increases in flash droughts in spring, but considering that only the SPI and SPEI reported a statistically significant trend, these increases could relate to variations in precipitation.

In addition to the markedly seasonal character indicating the role of AED in the development of flash droughts, we also found a strong spatial component determining the role of AED in these events in Spain. The AED shows strong spatial differences in Spain, with a clear north–south gradient between the humid regions of the north and the drier regions of southern and central Spain [36,39]. Several studies suggested that the contribution of AED is much higher in dry regions than in humid ones, since its role is only important during periods of low precipitation or limited soil moisture [9,17]. This could explain the contrasts found between the drier regions of central and southern Spain and the humid ones in northern Spain, which, even in summer, exhibited very low sensitivity to AED. Thus, during summer, when the AED in Spain is high and precipitation is generally very low [40], strong AED anomalies play a key role in the development of flash droughts, especially in dry areas such as the Ebro Depression and southern Spain. Although the increased in AED was probably the main driver of changes in flash drought frequency in summer, we must stress that we have demonstrated that precipitation is still the main driver in the temporal variability of droughts in Spain, and it is also the main contributing factor in triggering flash droughts. Precipitation deficits are the major climatic variable triggering drought conditions [66,67], and it seems reasonable to expect that precipitation also has an essential role in the development of flash droughts. We have demonstrated this pattern in winter as well as in spring and autumn. However, it is possible that the AED has a relevant role in the development of certain flash drought events in the late spring.

The results of this research cannot be easily extrapolated. Spain is characterized by high spatial and seasonal variability in precipitation and AED, so it was expected that

their role in the development of flash droughts would be strongly variable. However, the role of precipitation and AED anomalies may exhibit significant changes in other regions of the world, and flash droughts could develop under diverse conditions. For example, Hobbins et al. [52] and Pendergrass et al. [19] pointed out that the occurrence of flash droughts is typically related to a change from energy-limited to water-limited conditions. This can be true in some cases, but it cannot be considered as a situation characteristic for most flash drought events worldwide. At a global scale, previous studies found similar patterns in the contribution of AED during periods characterized by low levels of precipitation in sub-humid regions [9]. Specifically, certain studies also indicated that precipitation is the main variable explaining the occurrence of flash droughts in the United States [63]. However, considering the complexity of the role played by AED in drought development, and its spatial and seasonal variability [17], further research into these issues is needed.

## 5. Conclusions

This study focused on the analysis of the role of precipitation and atmospheric evaporative demand (AED) in the occurrence of flash droughts in Spain. For this purpose, we analyzed the spatial and temporal patterns of flash drought identified through different drought indices based on precipitation (SPI), AED (EDDI), and both (SPEI). We also examined the sensitivity of the SPEI to AED to clarify the possible contribution of precipitation deficits and anomalous increases in AED to the development of flash droughts. The main conclusions from this study are as follows:

- Standardized drought indices such as SPI, EDDI, and SPEI are robust metrics for the identification of flash droughts. However, the use of indices based exclusively on precipitation or AED may have some limitations under certain circumstances.
- The spatial and temporal patterns of flash droughts can be highly variable, depending on the metrics used in analysis.
- Flash droughts in Spain can be triggered by both precipitation deficits and increases in AED, but their contribution to the development of flash droughts is highly variable spatially and seasonally.
- Precipitation is the main variable driving flash droughts in Spain, although AED anomalies can play a crucial role in the development of some flash drought events, especially in arid areas during the warm season.
- The sensitivity of the SPEI to AED during dry periods enables the drought conditions triggered by anomalous decreases of precipitation and/or increases of the AED to be captured, making it possible to identify and characterize flash droughts over very different climatic conditions seasonally and spatially.

**Supplementary Materials:** The following are available online at https://www.mdpi.com/2073-4433/12/2/165/s1, Figure S1: Temporal evolution of the annual and seasonal differences (events/for each grid point) between the flash drought series recorded by the SPI, EDDI and SPEI on mainland Spain and the Balearic Islands over the period 1961–2018, Figure S2: Monthly frequency of the sensitivity (%) of the SPEI to AED on mainland Spain and the Balearic Islands over the period 1961–2018 at a short time scale (1-month). The monthly series include the weekly data for the last week of each month in each year.

**Author Contributions:** Data curation, I.N., F.D.-C. and S.M.V.-S; Formal analysis, I.N., F.D.-C. and S.M.V.-S.; Funding acquisition, S.M.V.-S.; Software, I.N.; Supervision, I.N., F.D.-C. and S.M.V.-S.; Writing—original draft, I.N., F.D.-C. and S.M.V.-S; Writing—review & editing, I.N., F.D.-C. and S.M.V.-S. All authors have read and agreed to the published version of the manuscript.

**Funding:** Research projects CGL2017-82216-R, PCI2019-103631, and PID2019-108589RA-I00 financed by the Spanish Commission of Science and Technology and FEDER; CROSSDRO project financed by the AXIS (Assessment of Cross(X)—sectoral climate Impacts and pathways for Sustainable transformation), JPI-Climate co-funded call of the European Commission and INDECIS which is part

of ERA4CS, an ERA-NET initiated by JPI Climate, and funded by FORMAS (SE), DLR (DE), BMWFW (AT), IFD (DK), MINECO (ES), ANR (FR) with co-funding by the European Union (Grant 690462).

**Institutional Review Board Statement:** Not applicable.

**Informed Consent Statement:** Not applicable.

**Data Availability Statement:** Data used in this study is available under request.

**Acknowledgments:** The first author would like to thank the predoctoral fellowship "Subvenciones destinadas a la contratación de personal investigador predoctoral en formación (2017)" awarded by the Gobierno de Aragón (Government of Aragón, Spain).

**Conflicts of Interest:** The authors declare no conflict of interest.

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
