# Peer review of "Flash Drought Response to Precipitation and Atmospheric Evaporative Demand in Spain"

_atmosphere, doi:10.3390/atmos12020165_

Round 1

Reviewer 1 Report

The authors provide an assessment of flash drought in Spain and the Balearic Islands using three standardized drought indices.  The analysis focuses on the seasonal characteristics of flash drought occurrence, trends, and mechanisms over a recent 60-year period.  Overall, the manuscript is thorough, the methods are sound, conclusions are supported by the analysis, and the results/discussion provide physical insight with regard to flash drought over this region and more generally.  I think the paper would be of interest to the flash drought research community, as well as those interested specifically in the local climate of this region.  Nonetheless, I do have comments (generally minor) that I feel should be addressed prior to publication of the manuscript.  As seen below, my comments mainly relate to clarification of methods, physical interpretation of results, and presentation.

Comments:

L51-58: A relevant reference that fits into this discussion is Osman et al. (2020), and I suggest the authors include this citation here (see reference at the end of this review).

It is unclear what the following statement (appearing in many figure captions) means: “The monthly series include the weekly data for the last week of each month in each year.”  This seems to suggest that just one week of data from each month is used (i.e., a 7-day average at the end of each month, with the rest of the month ignored or discarded).  Is this the correct interpretation?  If so, the sensitivity of the results to this methodological choice is called into question.  In particular, a single week is likely to have high variability (interannually and within one month), potentially influencing the interpretation of the results.  For example, if you use the first week of each month (or second week, or third week) instead, would you get the same qualitative results?  Why not just average the values over the entire month prior to conducting the analysis?  Some additional clarification/justification is necessary here.

In the discussion of Figs. 2 and 8 (L165 and L347, respectively), and in their figure captions (L181 and L382, respectively), I feel the terminology “spatial correlation” is incorrectly used.  What is actually shown, assuming I am interpreting the results correctly, is the “spatial pattern” of the temporal correlation between indices (or in other words, the correlation between the time series of one index and that of another, shown at each grid point).  “Spatial correlation,” on the other hand, is what is actually depicted in Fig. 4.  Thus, I suggest rephrasing the descriptions of Figs. 2 and 8 using the terminology “spatial pattern of the correlation and associated significance…” or something similar.  

Fig. 5: Some of the areas of statistically significant trends are quite small, raising the possibility that they are occurring by chance.  Have you looked into the field significance of these trends?

Fig. 1 and Fig. 6: Please clarify in the figure caption that the regional mean is being shown.

Fig. 7: I am unsure of the value this figure, as it does not seem to add much beyond what can be deduced from Fig. 6 and by itself is hard to interpret (indeed, I found myself referring to Fig. 6 to interpret Fig. 7).  A suggestion is to remove this figure and only show Fig. 6, while placing a bit more emphasis in the text on the differences in Fig. 6 that you wish to highlight.  (Anyway, there appears to be a typo on L344 in the discussion of Fig. 7, where “SPEI” should be “EDDI”).

Fig. 10: The text on this figure is too small and is blurry.  I strongly suggest improving the quality and legibility of this figure, perhaps by rearranging the panels so that each panel can be bigger.

Discussion: With regard to the practical implications of flash drought and the interpretation of the important role of AED in inducing flash droughts in southern Spain in summer, it would be helpful to provide a bit more detail about the climate of this region in the text.  Exactly how dry is it during the summer under normal conditions in southern Spain?  Is there enough soil moisture to lose that flash drought would have a meaningful impact on agriculture and/or water resources, or is this region essentially always in “drought” during summer anyway?  Related to the physical mechanisms for AED, is it reasonable to hypothesize that the region begins in a water-limited evaporative regime, such that land-atmosphere feedbacks play a large in inducing flash drought?  Overall, I think some additional discussion on the above topics, along with more context about the climatology of Spain, would strengthen this aspect of the discussion.

Writing/typos:

Abstract, L11: Change “stress vegetation” to “stresses vegetation”

Abstract, L17-18: “The results show large differences between the spatial and temporal patterns of flash droughts recorded by each index.” – This statement is somewhat confusing.  I suggest rewording to “The results show large differences in the spatial and temporal patterns of flash droughts between indices.”

L103: “for each cell by grid point” – there may be a typo here.  Do you just mean “for each grid point”?

L214-216: This statement seems to be referring to spring, but the text says winter.

Fig. 5 caption, L273: Change “flash droughts events” to “flash drought events”

L288: “of the of” – delete second “of”

L305: The correlation value on the plot is “0.85” but it says “0.75” in the text.

L311-313: This statement appears to be about summer, but the text says spring.

Table 1: “Septembre”, “Octobre” -> “September”, “October”

L462-464: “we think that by implementing different standardized indices in monitoring and identifying them understanding of the behavior of this type of drought can be improved” – There appears to be a typo in this sentence, making the meaning unclear.

L522: “of a number” -> “of the number” ?

L534: “the most of study area” -> “most of the study area”

L542: “development flash droughts” -> “development of flash droughts”

L545: Should say “southern” rather than “northern”

References:

Osman et al. 2020, Flash drought onset over the Contiguous United States: Sensitivity of inventories and trends to quantitative definitions. Hydrology and Earth System Sciences. https://doi.org/10.5194/hess-2020-385

Author Response

Reviewer 1:

The authors provide an assessment of flash drought in Spain and the Balearic Islands using three standardized drought indices.  The analysis focuses on the seasonal characteristics of flash drought occurrence, trends, and mechanisms over a recent 60-year period.  Overall, the manuscript is thorough, the methods are sound, conclusions are supported by the analysis, and the results/discussion provide physical insight with regard to flash drought over this region and more generally.  I think the paper would be of interest to the flash drought research community, as well as those interested specifically in the local climate of this region.  Nonetheless, I do have comments (generally minor) that I feel should be addressed prior to publication of the manuscript.  As seen below, my comments mainly relate to clarification of methods, physical interpretation of results, and presentation.

Thanks for your positive assessment of the manuscript and the constructive comments. These comments have contributed significantly to improving the readability and clarity of the manuscript. Herein, we include a detailed response to all your comments.

Comments:

L51-58: A relevant reference that fits into this discussion is Osman et al. (2020), and I suggest the authors include this citation here (see reference at the end of this review).

The reference has been included, thank you. We think that this reference is very appropriate for the topic of this study.

It is unclear what the following statement (appearing in many figure captions) means: “The monthly series include the weekly data for the last week of each month in each year.”  This seems to suggest that just one week of data from each month is used (i.e., a 7-day average at the end of each month, with the rest of the month ignored or discarded).  Is this the correct interpretation?  If so, the sensitivity of the results to this methodological choice is called into question.  In particular, a single week is likely to have high variability (interannually and within one month), potentially influencing the interpretation of the results.  For example, if you use the first week of each month (or second week, or third week) instead, would you get the same qualitative results?  Why not just average the values over the entire month prior to conducting the analysis?  Some additional clarification/justification is necessary here.

Here, there are two important points to consider; on the one hand the frequency of the data (weekly), and on the other hand the temporal frequency of the index (1-month). Each weekly data used to calculate drought indices is accumulated over four weeks, which correspond to a 1-month time scale. Therefore, the weekly data for the last week of each month reflects the variability of the whole month.

Nonetheless, we understand the concern by the reviewer. This was detailed in the section 2.4 of the revised manuscript: “Given that each weekly data is calculated based on the precipitation, ETo or climate balance (P-ETo) data accumulated over four weeks (corresponding to 1-month time scale), we used the weekly data for the last week of each month to calculate the correlation between indices in order to reflect the variability over the whole month.”

In the discussion of Figs. 2 and 8 (L165 and L347, respectively), and in their figure captions (L181 and L382, respectively), I feel the terminology “spatial correlation” is incorrectly used.  What is actually shown, assuming I am interpreting the results correctly, is the “spatial pattern” of the temporal correlation between indices (or in other words, the correlation between the time series of one index and that of another, shown at each grid point).  “Spatial correlation,” on the other hand, is what is actually depicted in Fig. 4.  Thus, I suggest rephrasing the descriptions of Figs. 2 and 8 using the terminology “spatial pattern of the correlation and associated significance…” or something similar.  

We agree. As suggested by reviewer 1, we have replaced “Spatial correlation (Pearson’s r) and significance” by “Spatial pattern of the correlation (Pearson’s r) and associated significance”.

Fig. 5: Some of the areas of statistically significant trends are quite small, raising the possibility that they are occurring by chance.  Have you looked into the field significance of these trends?

We do not think this can occur by chance as the spatial patterns are very robust, with clear gradient from areas with positive to negative correlations.

Fig. 1 and Fig. 6: Please clarify in the figure caption that the regional mean is being shown.

Detailed in the revised manuscript.

Fig. 7: I am unsure of the value this figure, as it does not seem to add much beyond what can be deduced from Fig. 6 and by itself is hard to interpret (indeed, I found myself referring to Fig. 6 to interpret Fig. 7).  A suggestion is to remove this figure and only show Fig. 6, while placing a bit more emphasis in the text on the differences in Fig. 6 that you wish to highlight.  (Anyway, there appears to be a typo on L344 in the discussion of Fig. 7, where “SPEI” should be “EDDI”).

Given the high variability of the flash drought series (Fig. 6), we think that the inclusion of the Fig. 7 can be useful to illustrate more clearly the evolution of the differences between the series and their associated statistical significance. Nonetheless, we included this figure as supplementary material (Figure S1) in the revised manuscript and modified the text to emphasize the most relevant results.

L-344 “SPEI” was replaced by “EDDI”, thanks you.

Fig. 10: The text on this figure is too small and is blurry.  I strongly suggest improving the quality and legibility of this figure, perhaps by rearranging the panels so that each panel can be bigger.

 Modified in the revised manuscript.

Discussion: With regard to the practical implications of flash drought and the interpretation of the important role of AED in inducing flash droughts in southern Spain in summer, it would be helpful to provide a bit more detail about the climate of this region in the text.  Exactly how dry is it during the summer under normal conditions in southern Spain?  Is there enough soil moisture to lose that flash drought would have a meaningful impact on agriculture and/or water resources, or is this region essentially always in “drought” during summer anyway?  Related to the physical mechanisms for AED, is it reasonable to hypothesize that the region begins in a water-limited evaporative regime, such that land-atmosphere feedbacks play a large in inducing flash drought?  Overall, I think some additional discussion on the above topics, along with more context about the climatology of Spain, would strengthen this aspect of the discussion.

During summer the precipitation in southern regions of Spain is very low (generally close to zero) and periods characterized by low soil moisture are common. We understand the reviewer’s point, but we have to consider (given that we using AED as metric) that under water-limited conditions, the increase of the AED would likely increase vegetation stress for example by the inability to photosynthesize because the atmosphere is too dry for stomata to open. This issue would reduce carbon uptake  and cause relevant agricultural and environmental impacts. So it seems reasonable to expect that anomalous increases in AED (caused for example by heat waves and maybe reinforced by land–atmosphere feedbacks) can be a driver of agricultural droughts.

Nonetheless, it is not an issue free of discussion in the scientific community. Some studies such as Vicente-Serrano et al. 2020 or Miralles et al. 2019 have analyzed this topic in more depth.

We included additional information about the climate of Spain in introduction section.

Writing/typos:

Abstract, L11: Change “stress vegetation” to “stresses vegetation”

We agree. As suggested by reviewer 1, we have replaced “stress vegetation” by “stresses vegetation

Abstract, L17-18: “The results show large differences between the spatial and temporal patterns of flash droughts recorded by each index.” – This statement is somewhat confusing.  I suggest rewording to “The results show large differences in the spatial and temporal patterns of flash droughts between indices.”

We agree. As suggested by reviewer 1, we have replaced “The results show large differences between the spatial and temporal patterns of flash droughts recorded by each index.” by “The results show large differences in the spatial and temporal patterns of flash droughts between indices.”

L103: “for each cell by grid point” – there may be a typo here.  Do you just mean “for each grid point”?

For each grid point is correct. Thanks for the suggestion.

L214-216: This statement seems to be referring to spring, but the text says winter.

The reviewer is right, we have replaced “winter” by “spring”.

Fig. 5 caption, L273: Change “flash droughts events” to “flash drought events”

Replaced. Thanks.

L288: “of the of” – delete second “of”

Deleted. Thanks.

L305: The correlation value on the plot is “0.85” but it says “0.75” in the text.

Replaced. Thanks.

L311-313: This statement appears to be about summer, but the text says spring.

Replaced. Thanks.

Table 1: “Septembre”, “Octobre” -> “September”, “October”

 Replaced. Thanks.

L462-464: “we think that by implementing different standardized indices in monitoring and identifying them understanding of the behavior of this type of drought can be improved” – There appears to be a typo in this sentence, making the meaning unclear.

The sentence has been rewritten as follows: “we think that the implementation of different indices for flash drought analysis can be useful for a more comprehensive understanding”.

L522: “of a number” -> “of the number” ?

 Replaced. Thanks.

L534: “the most of study area” -> “most of the study area”

Deleted. Thanks.

L542: “development flash droughts” -> “development of flash droughts”

Replaced. Thanks.

L545: Should say “southern” rather than “northern”

The reviewer is right, we have replaced “northern” by “southern”.

Reviewer 2 Report

The manuscript entitled “Flash Drought Response to Precipitation and Atmospheric Evaporative Demand in Spain” is a research article comparing different drought indices based on precipitation or/and atmospheric evaporative demand in mainland Spain and the Balearic islands. The authors used 1-month timescale for conducting a spatiotemporal analysis by using gridded data of the time period 1961-2018,, in order to identify flash droughts.

The assessment of the performance of different drought indices in western Mediterranean could be quite useful both for the climate-meteorology researchers and for the Stakeholders as well.

The work is generally well organized and presented. However, I have some comments and suggestions that, in my opinion, can enhance the quality of the paper.

Comments and Suggestions

  1. The title does not exactly reflect the content of the manuscript. A more appropriate title could be “Use and relationships of indices based on precipitation and atmospheric evaporative demand to detect flash droughts in Spain” or “Comparison (or Assessment) of the SPI, SPED and EDDI standardized indices for identifying flash droughts in Spain”
  2. Editing of the text is necessary. Try to avoid unnecessary repetition (e.g. lines 78, 117, 150-151, 185-186, 259-260,293-294, 321-322, 387-388, etc).
  3. I would suggest to make some expression changes in order to make your text more easy to read. For example, I recommend to avoid presenting your results as “Figure 1 shows….”, “Figure 2 shows ….”, etc. You can change lines 149-151 from “Figure 1 shows….. variability in all months” to “In general, indices present high interannual variability in all months (Fig. 1)”, etc
  4. In line 31 is stated that “drought is considered to be a climate phenomenon”. Better “meteorological phenomenon”.
  5. Line 41. Use “factor” instead of “variable”
  6. Line 52. Use “acceptable” instead of “accepted”
  7. Line 54. Use “soil moisture” instead of “soil moisture data”
  8. Rephrase line 59-60. Suggestion “In this work we studied the impacts of precipitation deficit and atmospheric evaporative demand increase on flash droughts in Spain.”
  9. Line 64. Better “temporal variation” than “temporal behavior”
  10. Line 65. Better “them” than “these”
  11. Line 65. “enable” not “enables”
  12. Line 82. Give some additional information about the gridded data, their quality control, the interpolation methods used and the possible uncertainties, in a brief text to help the reader.
  13. Sub-section 2.2. A little more expanded description of the drought indices would be much helpful, especially for the EDDI and also for the AED.
  14. Lines 104, 107. Give more details concerning the study of Noguera et al. to help the readers and describe in a couple of lines Tomas-Burguera et al. methodology (line 144).
  15. Lines 136-140 can be deleted.
  16. Lines 152-160. The correlation analysis presented here is useful. However it is not directly associated with Figure1. A couple of lines describing the extreme drought events (extreme values of the drought indices presented in Figure 1) would be useful. Probably a short discussion should also be added in the Discussion section, concerning these strong droughts.
  17. In Figure 1 rearrange the graphs from January to December.
  18. All figures need editing. Make sure that all graphs are in a good size to be printed correctly. Unfortunately, most of the figures are not easy to read. Please check also the necessity for giving so many graphs. I believe that many of them can be given as supplemental material e.g. Trend significance in Fig. 5, Figure7, Fig. 9a etc
  19. Figures 3 and 4. A vertical (per index) presentation of the graphs will give horizontal space to make the graphs bigger and more easy to read.
  20. Lines 247-248. I think that it should be transferred to the materials and methods section along with a brief description of the “Monte Carlo approach” mentioned here.
  21. Line 272. Please check the expression “change per decade” in conjunction with what is depicted the figure.
  22. Lines 307-309. Please rephrase the sentence.
  23. Figure 6. Please check the p values for winter.
  24. In Table 1 would be useful to add the Precipitation sensitivity as well and make a short discussion.
  25. Lines 407-411, 413-415, 418-419 could better be part of the Discussion section. Also try to add some references.
  26. Line 425. Change the term “indisputable”
  27. Line 591. Use “arid” instead of “dry” areas.
  28. It seems that region’s aridity seriously affect the ability of the tested drought indices to detect flash drought. This is repeatedly found at specific areas in the Spanish peninsula according to your findings (e.g. the Med coastline). I would recommend to add a short description for the Region’s climate and its classification in the materials and methods section. Also some more and specific additions concerning the historical droughts in Spain would be useful in the introduction section.

Author Response

Reviewer 2:

The manuscript entitled “Flash Drought Response to Precipitation and Atmospheric Evaporative Demand in Spain” is a research article comparing different drought indices based on precipitation or/and atmospheric evaporative demand in mainland Spain and the Balearic islands. The authors used 1-month timescale for conducting a spatiotemporal analysis by using gridded data of the time period 1961-2018, in order to identify flash droughts.

The assessment of the performance of different drought indices in western Mediterranean could be quite useful both for the climate-meteorology researchers and for the Stakeholders as well.

The work is generally well organized and presented. However, I have some comments and suggestions that, in my opinion, can enhance the quality of the paper.

We are very grateful for the positive consideration of the results presented in the manuscript. We have carefully followed the suggestions and comments proposed by the reviewer.

Comments and Suggestions

The title does not exactly reflect the content of the manuscript. A more appropriate title could be “Use and relationships of indices based on precipitation and atmospheric evaporative demand to detect flash droughts in Spain” or “Comparison (or Assessment) of the SPI, SPEI and EDDI standardized indices for identifying flash droughts in Spain”

We think that the suggested titles do not reflect the entire scope of this study. The analyses presented in this research are focused on examining the response of flash droughts to anomalous changes in precipitation and AED. We think that the comparison between indices based on different variables (precipitation, ET0 and the difference between precipitation and ETo) as indicators of flash drought conditions, as well as the sensitivity analysis calculated by means of the SPEI, enables to unravel the role of precipitation deficits and the anomalous increase in AED in the occurrence of flash droughts.

Editing of the text is necessary. Try to avoid unnecessary repetition (e.g. lines 78, 117, 150-151, 185-186, 259-260,293-294, 321-322, 387-388, etc).

We have been modified the text in the revised manuscript, removing unnecessarily repeated sentences (see also next point of the review):

L 116-117. Deleted “on mainland of Spain and the Balearic Islands over the period 1961-2018.

L 184-185. Deleted “on mainland Spain and the Balearic Islands

L 260. Deleted “on mainland Spain and the Balearic Islands

I would suggest to make some expression changes in order to make your text more easy to read. For example, I recommend to avoid presenting your results as “Figure 1 shows….”, “Figure 2 shows ….”, etc. You can change lines 149-151 from “Figure 1 shows….. variability in all months” to “In general, indices present high interannual variability in all months (Fig. 1)”, etc

Numerous expressions of the text have been modified in the revised manuscript:

L 149-151 Modified “In general, the indices present high interannual variability in all months (Figure 1).

L 165-166 Modified “The spatial pattern of the correlation and associated significance between monthly series of the SPI, EDDI and SPEI over the period 1961-2018 also show some relevant differences (Figure 2)

L 235-239 Modified “In general, there was non-significant spatial correlation between the total number of events identified for each index and the shared variance was not high, due to notable spatial differences annually and seasonally (Figure 4).

L 258 “shows” was replaced by “depicts

L 292-295 Modified “The temporal evolution of the average frequency of flash drought events identified by the SPI, EDDI and SPEI on mainland Spain and the Balearic Islands over the period 1961-2018 showed a high variability at annual and seasonal scales (Figure 6). The annual series of flash droughts recorded by the SPI, EDDI and SPEI exhibited statistically significant correlations among them,…”

L 320-324 Modified “The difference in the number of flash droughts recorded between the SPI and EDDI, and also between the SPI and SPEI, showed a significant decrease over the period 1961-2018 at annual scale (Figure S1).

L 398-399 Modified “The monthly frequency and spatial distribution of the sensitivity of SPEI to AED from 1961-2018 at a 1-month time scale also presents notable contrasts (Figure 8).”

L 420 “shows” was replaced by “illustrates

In line 31 is stated that “drought is considered to be a climate phenomenon”. Better “meteorological phenomenon”.

We agree. As suggested by reviewer 2, we have replaced “climate phenomenon” by “meteorological phenomenon

Line 41. Use “factor” instead of “variable”

 Replaced. Thanks.

Line 52. Use “acceptable” instead of “accepted”

Replaced. Thanks.

Line 54. Use “soil moisture” instead of “soil moisture data”

We deleted “data

Rephrase line 59-60. Suggestion “In this work we studied the impacts of precipitation deficit and atmospheric evaporative demand increase on flash droughts in Spain.”

The sentence has been rewritten as follows: “In this work we studied the role of precipitation deficit and atmospheric evaporative demand increase on flash droughts in Spain,

Line 64. Better “temporal variation” than “temporal behavior”

Replaced. Thanks.

Line 65. Better “them” than “these”

Replaced. Thanks.

Line 65. “enable” not “enables”

Replaced. Thanks.

Line 82. Give some additional information about the gridded data, their quality control, the interpolation methods used and the possible uncertainties, in a brief text to help the reader.

We include more details about gridded dataset, quality control and interpolation methods in the revised manuscript:

The gridded dataset was created based on all daily observational information from the National Spanish Meteorological Service (AEMET) by means of an interpolation scheme of universal kriging using as input the meteorological data measured in the different meteorological stations and the terrain elevation. The climate series were subjected to homogenization process and a careful quality control (see details in Tomas-Burguera et al. [46]). Additional information about the dataset development, interpolation methodology and validation are available in Vicente-Serrano et al. [47].

Sub-section 2.2. A little more expanded description of the drought indices would be much helpful, especially for the EDDI and also for the AED.

Brief details were included in the revised manuscript.

Lines 104, 107. Give more details concerning the study of Noguera et al. to help the readers and describe in a couple of lines Tomas-Burguera et al. methodology (line 144).

More details about Noguera et al. 2020 are already described in 2.3 section: “The original approach was based on the SPEI at a short time scale (1-month) and high-frequency data (weekly) to identify the onset of flash drought episodes. This method focuses on the rapid development characteristic of flash droughts, which results in a sudden, very sharp drop in the index values. Thus, flash drought is defined as..”

We included a brief description of Tomas-Burguera et al. 2020 methodology in 2.5 section: “Thus, using the precipitation and AED series employed to compute SPEI, we calculated the partial derivatives of the climate balance (D) to determinate the relative contribution of both variables in each month over the period 1961-2018. The series of precipitation and AED were detrended prior to making the analysis to avoid the possible effects of trends on the results (see more details in Tomas-Burguera et al. [9]).”

Lines 136-140 can be deleted.

We think that this is relevant information to understand more clearly the process followed in the analysis of sensitivity of SPEI to AED and its comparison with the total frequency of flash drought events.

Lines 152-160. The correlation analysis presented here is useful. However it is not directly associated with Figure1. A couple of lines describing the extreme drought events (extreme values of the drought indices presented in Figure 1) would be useful. Probably a short discussion should also be added in the Discussion section, concerning these strong droughts.

We understand the reviewer’s interest in the drought periods shown in regional series of the Figure 1. Nonetheless, these episodes do not consider the rapid development characteristic of flash droughts and its description and characterization would require a specific analysis, which is not the objective of this research. The objective of this figure is exclusively to compare the behavior and spatial variability of the indices, since this is a key point to interpret the results obtained by each index in the identification of flash drought.

In Figure 1 rearrange the graphs from January to December.

Modified in the revised manuscript

All figures need editing. Make sure that all graphs are in a good size to be printed correctly. Unfortunately, most of the figures are not easy to read. Please check also the necessity for giving so many graphs. I believe that many of them can be given as supplemental material e.g. Trend significance in Fig. 5, Figure7, Fig. 9a etc

All figures were edited in the revised manuscript. Figure 7 and 9a were included as supplementary material (Figure S1 and Figure S2, respectively).

Figures 3 and 4. A vertical (per index) presentation of the graphs will give horizontal space to make the graphs bigger and more easy to read.

Modified in the revised manuscript

Lines 247-248. I think that it should be transferred to the materials and methods section along with a brief description of the “Monte Carlo approach” mentioned here.

Deleted of the figure caption and detailed in the revised manuscript in 2.4 section:

“In addition, we analyzed the relationship between the total frequency of flash droughts identified by each index and the associated significance. The significance of Pearson’s r coefficients was estimated using a Monte Carlo approach, in which the total frequency of flash droughts recorded by SPI, EDDI and SPEI was correlated in 1000 random samples of 30 points from the entire dataset at annual and seasonal scales.”

Line 272. Please check the expression “change per decade” in conjunction with what is depicted the figure.

This is correct.

Lines 307-309. Please rephrase the sentence.

Modified as follows: “The flash drought series also showed significant correlation in this season, the highest correlation was found between the SPI and SPEI (Pearson’s r = 0.84)

Figure 6. Please check the p values for winter.

These are correct.

In Table 1 would be useful to add the Precipitation sensitivity as well and make a short discussion.

As results are in % this is simply the difference to 100.

Lines 407-411, 413-415, 418-419 could better be part of the Discussion section. Also try to add some references.

Modified and included in the discussion in in the revised manuscript.

Line 425. Change the term “indisputable”

Modified in the revised manuscript as follows: “, so the role of precipitation in the development of flash droughts in those months seems clearly dominant

Line 591. Use “arid” instead of “dry” areas.

Replaced. Thanks.

It seems that region’s aridity seriously affect the ability of the tested drought indices to detect flash drought. This is repeatedly found at specific areas in the Spanish peninsula according to your findings (e.g. the Med coastline). I would recommend to add a short description for the Region’s climate and its classification in the materials and methods section. Also some more and specific additions concerning the historical droughts in Spain would be useful in the introduction section.

Additional information about the climate of Spain and historical droughts were included in the introduction section:

Several studies showed the high occurrence of severe droughts episodes in Spain through historical [28–30] and instrumental records [31]. Spain has strong climatic contrasts with marked spatial and seasonal differences in precipitation [32–35] and AED [36–39]. Northern Spain is characterized by its humid oceanic climate with abundant precipitation over the year, while northeastern and southeastern regions are mainly characterized by semiarid conditions with annual precipitation generally below 300 mm [40]. On the other hand, the complex topography given by the presence of numerous mountain chains, results in a strong continental features in central Spain. This climatic complexity..

References:

Tomas-Burguera, M.; Vicente-Serrano, S.M.; Peña-Angulo, D.; Domínguez-Castro, F.; Noguera, I.; El Kenawy, A. Global Characterization of the Varying Responses of the Standardized Precipitation Evapotranspiration Index to Atmospheric Evaporative Demand. J. Geophys. Res. Atmos. 2020, 125, doi:10.1029/2020JD033017.

Miralles, D.; Gentine, P.; Seneviratne, S.I.; Teuling, A.J. Land-atmospheric feedbacks during droughts and heatwaves: state of the science and current challenges. Ann. N. Y. Acad. Sci. 2019, 8, 469.

Vicente-Serrano, S.M.; McVicar, T.; Miralles, D.; Yang, Y.; Tomas-Burguera, M. Unravelling the influence of atmospheric evaporative demand on drought under climate dynamics. Wiley Interdiscip. Rev. Clim. Chang. 2020, in press.